# UI2V-Bench: An Understanding-based Image-to-video Generation Benchmark

## Abstract

Generative diffusion models are developing rapidly and attracting increasing attention due to their wide range of applications. Image-to-Video (I2V) generation has become a major focus in the field of video synthesis. However, existing evaluation benchmarks primarily focus on aspects such as video quality and temporal consistency, while largely overlooking the model's ability to understand the semantics of specific subjects in the input image or to ensure that the generated video aligns with physical laws and human commonsense. To address this gap, we propose UI2V-Bench, a novel benchmark for evaluating I2V models with a focus on semantic understanding and reasoning. It introduces four primary evaluation dimensions: spatial understanding, attribute binding, category understanding, and reasoning. To assess these dimensions, we design two evaluation methods based on Multimodal Large Language Models (MLLMs): an instance-level pipeline for fine-grained semantic understanding, and a feedback-based reasoning pipeline that enables step-by-step causal assessment for more accurate evaluation. UI2V-Bench includes approximately 500 carefully constructed text–image pairs and evaluates a range of both open-source and closed-source I2V models across all defined dimensions. We further incorporate human evaluations, which show strong alignment with the proposed MLLM-based metrics. Overall, UI2V-Bench fills a critical gap in I2V evaluation by emphasizing semantic comprehension and reasoning ability, offering a robust framework and dataset to support future research and model development in the field.

## 1 Introduction

Diffusion models Ho et al. (2020); Song et al. (2020); Dhariwal & Nichol (2021) have recently achieved remarkable success in image generation and have been rapidly extended to video generation. Early video diffusion models Wan et al. (2025); Kong et al. (2024); Yang et al. (2024) primarily focus on the text-to-video (T2V) task. However, in practical applications, the image-to-video (I2V) task is more prevalent and useful, as it takes both a textual prompt and an image as inputs. This not only enhances controllability but also makes the generation process more intuitive for users. With growing demand, I2V models have proliferated rapidly, becoming a major focus of current video generation research.

Despite this progress, evaluation metrics for the I2V task remain limited. Existing benchmarks, such as AIGCBench Fan et al. (2024) and AnimateBench Zhang et al. (2024), primarily focus on video quality, temporal consistency, and motion smoothness. However, since I2V models take an image as an additional input, they must not only produce visually high-quality videos but also accurately interpret the semantic content of the input image—an evaluation aspect often overlooked in current benchmarks. Beyond semantic comprehension, a model's reasoning ability is also crucial for generating logically coherent videos. It is important to assess whether the model can infer and synthesize events that conform to physical laws and common human understanding—for example, recognizing that *"pulling a trigger"* implies *"a bullet being fired"*. Overall, current evaluation protocols rarely assess such fine-grained semantics or implicit world knowledge, which limits their effectiveness in guiding meaningful model improvements.

To address these limitations, we propose UI2V-Bench, an understanding-based benchmark for evaluating image-to-video (I2V) generation models. This framework comprehensively assesses a model's

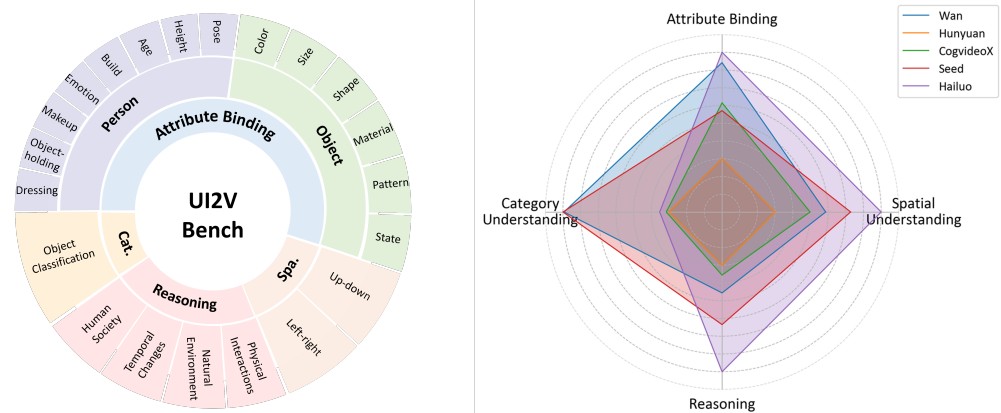

Figure 1: Overview of UI2V-Bench. Our understanding-based I2V benchmark consists of 4 dimensions: spatial understanding, attribute binding, category understanding and reasoning. We evaluate these four capabilities across five representative I2V models. Cat. represents category understanding dimension. Spa. represents spatial understanding.

ability to understand the semantic content of input images and to perform logical reasoning. It consists of four primary evaluation dimensions: spatial understanding, attribute binding, category understanding, and reasoning. As shown in Figure 1, these dimensions are further divided into 9 aspects and 19 fine-grained sub-dimensions. Spatial understanding examines how well the model perceives spatial relationships in the input image, such as left–right or up–down positioning. Attribute binding evaluates whether the model can correctly associate distinguishing attributes with specific subjects, including both people and objects. Category understanding measures the ability to differentiate among multiple object categories. Reasoning focuses on the model's capacity for causal inference across various domains, including human society, physical interactions, temporal changes, and the natural environment.

To evaluate these capabilities, we design two MLLM-based evaluation methods. For the three semantic understanding dimensions—spatial understanding, attribute binding, and category understanding—we develop an instance-level evaluation pipeline that supports fine-grained perceptual judgment, as illustrated in Figure 3. For the reasoning dimension, which is often overlooked in existing benchmarks, we propose a feedback-based pipeline that guides MLLMs through a chain of reasoning steps to ensure more accurate assessments, as shown in Figure 10.

In addition, the UI2V benchmark consists of approximately 500 carefully designed text–image pairs and evaluates a range of open-source models (e.g., Wan2.1 Wan et al. (2025), CogVideoX Yang et al. (2024), Hunyuan-video Kong et al. (2024)) and closed-source models (e.g., SeedDance[1] Gao et al. (2025),Hailuo[2]) across all defined dimensions. Human evaluators also assessed the generated videos, and the results show strong alignment between our evaluation scores and human judgments.

In summary, UI2V-Bench fills a critical gap in the evaluation of I2V models by emphasizing semantic and reasoning capabilities. It provides a systematic and reliable reference standard for guiding the optimization and improvement of future I2V models. We will release the dataset, evaluation suite to facilitate further research and development of I2V models in the community.

## 2 RELATED WORK

### 2.1 IMAGE-TO-VIDEO MODELS

In recent years, diffusion models(Ho et al. (2020);Song et al. (2020);Dhariwal & Nichol (2021);Ho et al. (2022)) have achieved remarkable progress in generative tasks.Early studies focused primarily on image generation(Nichol et al. (2021);Rombach et al. (2022);Hertz et al. (2022);Li et al.

---

[1]SeedDance 1.0 pro, https://www.volcengine.com/

[2]Hailuo2.0, https://hailuoai.com/

(2024);Kong et al. (2025)).Building on this, researchers extended diffusion models to video generation, particularly text-to-video (T2V)(Singer et al. (2022);Zhang et al. (2023);Wu et al. (2023);Chen et al. (2023);Zheng et al. (2024);Liu et al. (2024c)), which produces temporally coherent video content from natural language descriptions.However, many practical scenarios require not only text but often a single static image.This motivates the study of image-to-video (I2V) generation(HaCohen et al. (2024);Yang et al. (2024);Kong et al. (2024);Wan et al. (2025)), which typically incorporates both text and image conditions (TI2V) to preserve the appearance of the input image while introducing plausible temporal dynamics.Despite the rapid development of numerous I2V models, systematic benchmarks for comprehensive performance assessment remain lacking.

## 2.2 EVALUATION OF VIDEO GENERATION MODELS

Evaluation metrics for video generation generally cover visual quality and temporal consistency, with common measures including Inception Score (IS)(Salimans et al. (2016)) , Learned Perceptual Image Patch Similarity(LPIPS)(Zhang et al. (2018)), Fréchet Inception Distance (FID)(Heusel et al. (2017)) , Fréchet Video Distance (FVD)(Unterthiner et al. (2018)). In T2V tasks, several benchmarks(Liu et al. (2023);Huang et al. (2024a);Kou et al. (2024);Liu et al. (2024b);Wu et al. (2024);Sun et al. (2025)) have been proposed to systematically evaluate models along multiple dimensions. In contrast, benchmarks for I2V remain relatively limited.Existing studies such as AIGCBench(Fan et al. (2023)) and AnimateBench(Zhang et al. (2024)) mainly rely on CLIP-based scores, TC-Bench(Feng et al. (2024)) evaluates Temporal Compositionality, while VBenchI2V(Huang et al. (2024b)) assesses consistency by separating the input image and generated video into foreground and background.Nevertheless, these approaches largely overlook the central challenge of I2V: whether the model truly understands the input image and performs reasoning based on it.To bridge this gap, we introduce a new benchmark that provides systematic evaluation along four dimensions: spatial understanding, Category Understanding, attribute binding, and reasoning.

## 3 BENCH CONSTRUCTION

In this section, we introduce the main components of UI2VBench. Section 3.1 presents the four primary evaluation dimensions: Spatial Understanding, Attribute Binding, Category Understanding, and Reasoning, where the first three dimensions are together referred to as the semantic understanding dimension of instance-level evaluation. Section 3.2 then details the sources and construction methods of the input images and textual prompts.

### 3.1 DIMENSION SUITE

#### 3.1.1 SPATIAL UNDERSTANDING

Spatial understanding is a critical ability for I2V models. When multiple subjects of the same category appear in the input image with distinct spatial arrangements, users often specify one subject to animate by referring to its spatial location in the textual prompt. This requires the model to first comprehend the spatial relationships in the input image, then accurately localize the target subject, and finally animate it according to the prompt description.

In our benchmark, we construct the spatial understanding dimension using self-designed input cases to evaluate current mainstream I2V models. The spatial relationships considered primarily include vertical and horizontal arrangements. Specifically, when multiple subjects in an image are linearly arranged (e.g., left–right or up–down), the prompt identifies the target subject using spatial ordinal relationships. For example, for an image with two dogs aligned horizontally, a prompt could be: *"The dog on the left sits down, while the dog on the right remains still."*. To prevent spatial information leakage from category-level cues during subject localization, all subjects in this dimension are constrained to the same category. In addition, the number of subjects in our test cases ranges from two to four to enhance diversity and better simulating real-world scenarios.

#### 3.1.2 ATTRIBUTE BINDING

In images with multiple subjects, I2V models often need to animate a specific one, guided by textual descriptions of that target (e.g., *"the crying child"*, *"the black table"*). This dimension is designed to

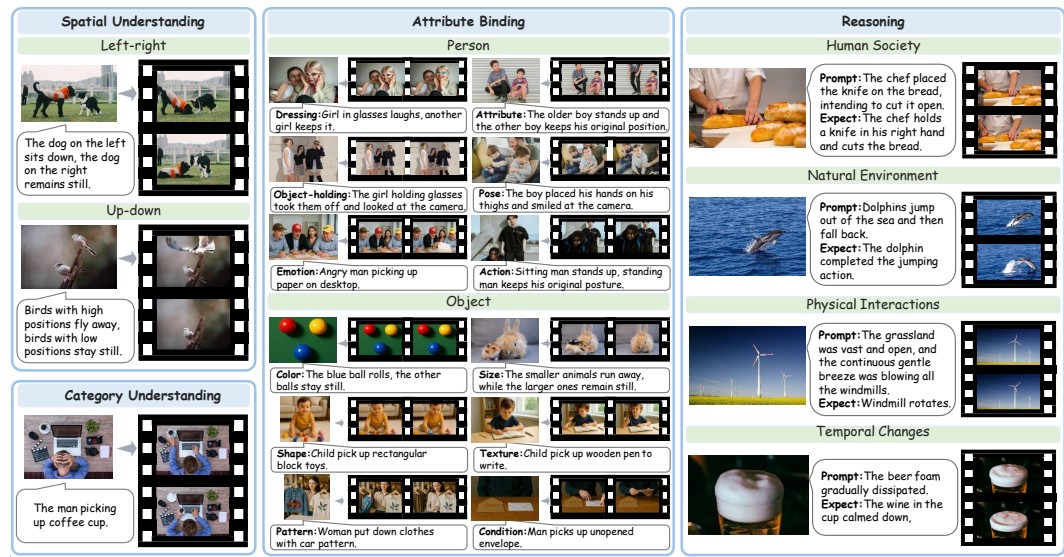

Figure 2: Generation cases across evaluation dimensions. For each text–image pair, we show the key video frames of the results corresponding to the 4 evaluation dimensions: spatial understanding, attribute binding, category understanding, and reasoning.

evaluate the model's ability to understand and utilize such attribute-based descriptions for accurate subject localization and animation. Overall, the cases are categorized into two primary types: **person animation** and **object animation**, each further divided into sub-dimensions based on attribute types. To ensure that models depend solely on attribute information rather than category differences, all subjects in each case are constrained to the same object category.

**Person animation.** Attributes may include age, height, build, dressing, makeup, emotion, pose, and object-holding. To specify the target subject based on age, height, or build, we employ comparative or superlative descriptors (e.g., *"the older person"*, *"the taller individual"*, or *"the heaviest"*). For dressing, the subject is identified by what they are wearing (e.g., *"the girl in a red dress"*). The makeup sub-dimension applies when the target person has distinctive cosmetics in the image (e.g., *"the girl with blue eyeshadow"*). For emotion, subjects may be described with expressions (e.g., *"a person with a happy expression"*). The pose sub-dimension includes descriptions such as *"the boy sitting on the ground"*. In the object-holding sub-dimension, the subject is identified by a unique item they are holding, such as *"the man holding a guitar"*.

**Object animation.** Specification can be achieved through attributes such as color, size, shape, material, pattern, and state. Color is a commonly used attribute for describing objects in images (e.g., *"the red balloon"*). Shape may include descriptors such as *triangular*, *cubic*, or *conical*. For the material, we construct cases with different materials such as *metal*, *plastic*, *wood*, and *glass*. Pattern is used when the object can be uniquely identified by surface markings (e.g., *"the building block with the letter A"*). Objects can also be distinguished by their state, for example, differentiating between *an empty cup* and *a cup containing water*.

### 3.1.3 CATEGORY UNDERSTANDING

The cases in the previous two dimensions— "Spatial Understanding" and "Attribute Binding"—are constructed under the constraint that all subjects belong to the same category. In contrast, the "Category Understanding" dimension is designed to assess whether I2V models can accurately perceive and distinguish multiple object categories present in a single input image. Typically, an image contains several common objects from different categories (e.g., *carrot*, *apple*, *banana*), and the model is required to animate the specified object(s) based on the textual prompt. This dimension evaluates not only the model's ability to recognize object categories but also its capacity to maintain identity consistency over time. For example, prompts like *"remove the carrot"* or *"the man picking up the*

*coffee cup"* require both precise localization of the target object and temporally consistent animation reflecting its dynamic state.

### 3.1.4 REASONING.

For the I2V generation task, a model's reasoning ability refers to its capacity to apply common-sense, logical, and physical knowledge in generating videos that are consistent with real-world dynamics and human interactions. Since existing video benchmarks rarely address this dimension, we introduce four sub-dimensions to comprehensively evaluate it: Human Society, Physical Interactions, Temporal Changes, and Natural Environment.

**Human Society.** This dimension evaluates the model's understanding and reasoning capabilities regarding human social behaviors, and relationship. We design representative test cases covering two key aspects: human intentio n reasoning and social relationship understanding. The former focuses on deducing the purpose of tool usage, behavioral goals, and underlying motivations (e.g. *"The archer released the bowstring"*) while the latter requires the model to identify and analyze professional associations, familial ties, or social roles between individuals(e.g. *"The older brother hugs his younger sister"*).

**Physical Interactions.** This dimension evaluates the model's ability to understand basic physical laws, focusing on object interactions and motion changes. A model with physical common sense should predict object dynamics under forces or environmental changes, ensuring that the generated videos follow real-world physical principles. Therefore, we design test cases covering core physical phenomena, including mechanics, fluid dynamics, and gravitational effects, to verify whether the model truly understands fundamental physics and avoids generating content that violates natural laws. (e.g. *"Let go of the balloon"*)

**Temporal Changes.** This dimension evaluates the model's ability to understand and predict how objects or scenes evolve over time. Beyond recognizing static attributes such as color, shape, or size, a model with strong reasoning skills should also infer how these attributes change through natural temporal progression. We design representative cases across different time scales, focusing on life processes, environmental changes, and material state transitions. These diverse scenarios comprehensively assess the model's capacity to reason about dynamic temporal changes (e.g. *"The banana was left outside for a week"*).

**Natural Environment.** This dimension evaluates the model's understanding of natural ecosystems and animal behavioral patterns. ensuring biological and ecological principles. A model with robust natural-world knowledge should accurately reason about the behaviors of animals and plants in specific environments. We collect a large number of images of various species of animals and plants, covering different types, including animal hunting, interactions between animals, plants adapting to environmental changes and so on. (e.g. *"The bee lands on the flower to collect nectar"*)

### 3.2 INPUT SUITE

I2V models require an image paired with a tailored text prompt as input to guide the video generation process. For the collection of input images, the majority are sourced from open-access websites such as Unsplash[3] and Pexels[4], with a small number of synthetic images generated by GPT-4 for some complex test scenes, such as *"Apples arranged in a circle"*.

Specifically, for the **Spatial Understanding** and **Attribute Binding** dimensions, where multiple subjects belong to the same category, we ensure that the prompt only drives one object or person, with the other subjects remaining stationary. For example, if an image contains three balls, the prompt would specify the blue ball, and only one ball would match this description. For the **Category Understanding** dimension, the input image contains various types of common objects, such as vegetables, fruits, and tools. For the **Reasoning** dimension, cases are designed to explore and reason about the hidden world knowledge behind the image. In this case, the prompts no longer describe a specific subject to drive its motion. Instead, they provide a precondition for the input image as the "cause," guiding the I2V model to generate the corresponding "effect."

---

[3]https://unsplash.com/
[4]https://pexels.com/

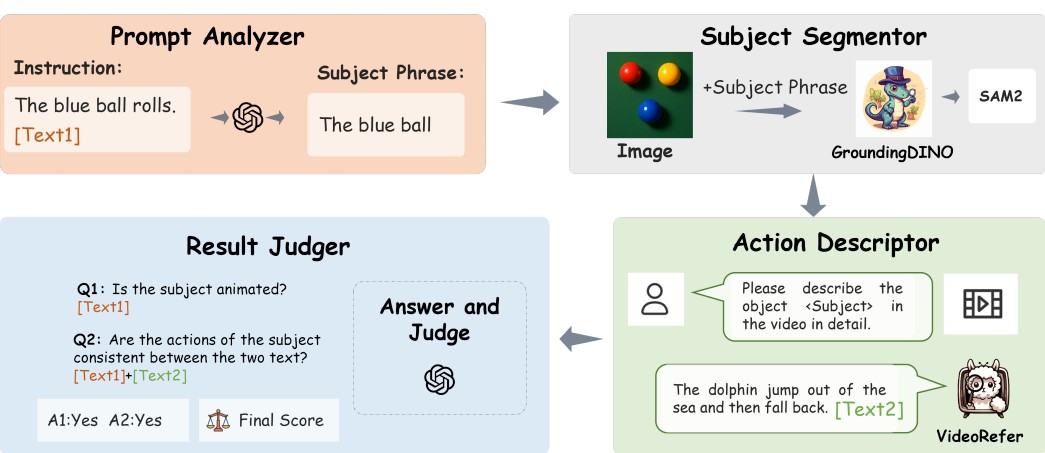

Figure 3: Proposed instance-level evaluation framework for semantic understanding. Given an input prompt and image, the framework first extracts the subject and action keywords, then obtains the target subject's mask, which is used to generate detailed descriptions. Finally, these descriptions are compared with the original prompt to produce the final evaluation scores.

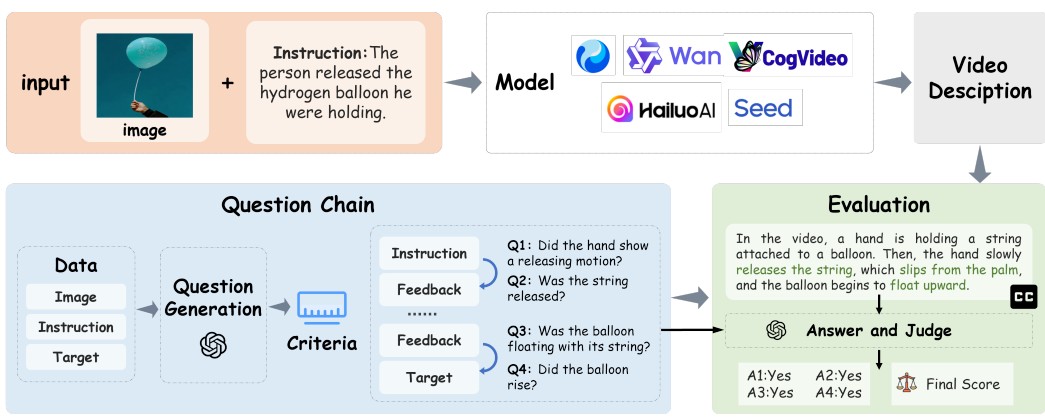

Figure 4: Proposed feedback-based evaluation framework for the Reasoning dimension. Given a textual prompt and generated video, the pipeline first produces a detailed video description, then generates a chain of intermediate, observable yes/no questions that guide step-by-step causal validation. Final and intermediate responses are jointly scored to assess the model's reasoning ability.

It is challenging to evaluate the image-understanding ability of I2V models, as it requires fine-grained and comprehensive cross-modal understanding. A straightforward yet reliable approach is to employ human evaluators who assess the generated results based on pre-defined rules for each evaluation dimension. However, this method is highly time-consuming and labor-intensive. Therefore, we design two evaluation pipelines based on MLLMs.

## 4  MLLM-BASED EVALUATION METRICS

### 4.1  INSTANCE-LEVEL EVALUATION FOR SEMATIC UNDERSTANDING

Current video generation evaluation methods primarily rely on general MLLMs (e.g., LLaVA, Qwen, GPT-4o) to assess output quality through multi-turn QA. However, we design an instance-level evaluation pipeline for three semantic understanding dimensions: Spatial Understanding, Attribute Binding, and Category Understanding, which similarly require the pipeline to possess fine-grained perceptual abilities.

As illustrated in Figure 3, the pipeline involves four components: (i) Prompt Analyzer; (ii) Subject Segmentor; (iii) Action Descriptor; (iv) Result Judge. Specifically, we use MLLMs as the Prompt Analyzer to first extract keywords related to the subject and action, which serve as key information for the subsequent workflow. The Subject Segmentor, composed of GroundingDINO Liu et al. (2024a) and SAM2 Ravi et al. (2024), takes subject keywords as input and extracts the target subject's mask from the input image. The obtained mask is then input into the Action Descriptor, which generates a detailed description of the action and the subject. The Action Descriptor utilizes an advanced spatial-temporal object understanding video-LLM VideoRefer Yuan et al. (2025b), rather than a common MLLM, which mainly focuses on general scene understanding. Finally, the Result Judge component utilizes an MLLM to output the final scores by comparing the subject description with the input prompt.

## 4.2 FEEDBACK-BASED EVALUATION FOR REASONING

The aforementioned fine-grained, instance-level evaluation methods are particularly suited for the dimensions of semantic understanding. For cases in the Reasoning dimension, greater attention should be given to reasoning about the underlying motivations of the image from the textual prompt input. In other words, while common evaluation pipelines focus more on instruction-following capabilities, the Reasoning dimension is primarily concerned with testing the model's ability for causal inference.

Evaluating reasoning capabilities faces two challenges. First, MLLMs struggle to accurately identify inconsistencies between text and video due to textual biases Han et al. (2025). Second, video content may not align with the target. For example, given an image of a hand holding a pistol and the instruction *"The sniper pulls the trigger"*, the generated video should show the bullet hitting the target, but a bullet is absent in the generated video due to its high speed.

As shown in Figure 10, we propose a novel feedback-based MLLM evaluation method that generates a chain of progressively validated questions, introducing intermediate feedback to help the MLLM correctly assess the quality of reasoning dimension evaluation cases. The specific steps are as follows:

1. Video Description: Generate a detailed description using a video description model (e.g., Tarsier Yuan et al. (2025a)) to represent the video content for subsequent evaluation.

2. Question Chain Generation: The LLM generates a set of questions based on preset text prompts and the target. Each question must be a binary judgment (yes/no) or an observable phenomenon that triggers a thought process leading to a feedback result. For example: *"Is there hand motion indicating the trigger is pulled?"*, *"Is there noticeable recoil from the gun?"*, *"Is there a flash of fire?"*, *"Determine if the bullet has been fired"*, etc.

3. Evaluation Score: Based on the detailed description generated in the first step and the chain of questions generated in the second step, assess the completion of the final result and the intermediate feedback, and output a score.

## 5 EXPERIMENTS

### 5.1 EXPERIMENTAL SETUP

**Evaluated models**. For a comprehensive evaluation of the current development in I2V generation, We adopt 3 mainstream open-source I2V models and 2 close-source commercial I2V models for evaluation. Specifically, the open-source models are Wan2.1(Wan et al. (2025)), Hunyuan-Video(Kong et al. (2024)), CogvideoX(Yang et al. (2024)). More will be added as they become open-sourced. Besides, we select 2 close-source commercial models: Seedance(Gao et al. (2025)), Hailuo, which gain awesome Elo score and high rank on Artificial Analysis Video Arena Leaderboard. The Arena Elo system, adapted from chess, objectively ranks models through anonymous user votes on randomized model matchups.

**Implementation details**. For each sub-ability dimension, videos are generated using the models based on the corresponding prompt suite described in Section 3.2. For the open-source models, we

Table 1: Benchmarking on Reasoning, spatial understanding, category understanding, and attribute binding.

| Model | Reasoning | | | | | Spatial Understanding | | | Category Understanding |
|---|---|---|---|---|---|---|---|---|---|
| | HS | NE | PI | TC | Avg. | Left-right | Up-down | Avg. | Object Classification |
| HunyuanVideo | 53.85 | 21.47 | 35.21 | 25.83 | 34.81 | 23.81 | 24.67 | 24.24 | 20.00 |
| CogvideoX | 52.18 | 35.90 | 32.50 | 27.67 | 37.80 | 30.78 | 36.4 | 33.59 | 20.26 |
| Wan2.1 | 53.51 | 48.98 | 41.46 | 24.92 | 43.18 | 41.72 | 33.80 | 37.76 | 30.04 |
| SeedDance | 73.45 | 50.77 | 45.91 | 38.25 | 52.77 | 44.04 | 44.77 | 44.41 | 30.15 |
| Hailuo | 71.14 | 71.58 | 62.76 | 70.19 | 67.04 | 49.5 | 55.64 | 52.57 | 20.89 |

| Model | Attribute Binding | | | | | | | | | | | |
|---|---|---|---|---|---|---|---|---|---|---|---|---|
| | Person | | | | | Object | | | | | | |
| | Dressing | Person-Attribute | Object-holding | Emotion | Person-Action | Color | Size | Shape | Material | Pattern | Condition | Avg. |
| HunyuanVideo | 24.26 | 27.50 | 23.33 | 20.00 | 21.67 | 23.23 | 56.67 | 51.67 | 56.30 | 27.92 | 51.46 | 34.91 |
| CogvideoX | 27.40 | 42.08 | 41.11 | 22.58 | 20.83 | 29.38 | 52.50 | 46.11 | 69.44 | 35.83 | 53.33 | 40.05 |
| Wan2.1 | 28.97 | 49.58 | 28.33 | 27.67 | 57.50 | 28.33 | 44.17 | 61.48 | 65.74 | 35.45 | 53.96 | 43.74 |
| SeedDance | 27.39 | 29.17 | 36.23 | 22.37 | 20.00 | 35.40 | 44.52 | 47.69 | 56.96 | 48.33 | 64.55 | 39.33 |
| Hailuo | 37.37 | 24.52 | 33.67 | 37.86 | 81.67 | 36.44 | 51.41 | 35.33 | 38.67 | 62.78 | 51.88 | 44.69 |

Table 2: Comparisons of video quality and video-condition alignment metrics.

| Model | Video Quality | | | Video-condition alignment | | |
|---|---|---|---|---|---|---|
| | Image Quality | Aesthetic Quality | Motion Smoothness | Video-text Alignment | Video-image Similarity | Image Understanding (Ours) |
| HunyuanVideo | 0.7066 | 0.6026 | 0.9941 | 0.2136 | 0.9305 | 28.49 |
| CogvideoX | 0.7097 | 0.5642 | 0.9887 | 0.2145 | 0.9229 | 32.93 |
| Wan2.1 | 0.7139 | 0.6024 | 0.9865 | 0.2093 | 0.9313 | 38.68 |
| SeedDance | 0.7321 | 0.6080 | 0.9926 | 0.1976 | 0.9009 | 41.67 |
| Hailuo | 0.7280 | 0.5909 | 0.9943 | 0.2048 | 0.8937 | 46.30 |

select the most advanced available version of each model for evaluation. For Wan2.1, we employ the 14B version, which yields 5-second videos rendered at 480p and 24 FPS.HunyuanVideo produces 3-second outputs at 720p resolution with a frame rate of 24 FPS. CogVideoX1.5-5B generates 6-second videos at a resolution of 768×1360 and 8 FPS. For the closed-source commercial models, we select versions that balance performance and API cost. Seedance-1.0-pro is employed to synthesize 5-second videos at 580p resolution and 25 FPS, while MinMax-Hailuo-02 is chosen for its superior generation quality, producing 6-second clips at 768p resolution and 24 FPS.

## 5.2 EVALUATION METRICS

**Our proposed Metrics**. In our experiments, we evaluate all models using four major metrics: Attribute Binding, Category Understanding, Reason, and Spatial Understanding, as shown in Table 1. HS denotes Human Society, NE denotes Natural Environment, PI denotes Physical Interactions, and TC denotes Temporal Changes. These metrics allow us to quantitatively compare model performance across different reasoning and spatial dimensions.

**Previous Metrics**. In addition, we measure several fundamental quality dimensions to ensure the comprehensiveness of our benchmark,as shown in Table 2 . Concretely, we report image quality, aesthetic quality and motion smoothness using VBench(Huang et al. (2024a)).These metrics complement our evaluation by focusing on essential aspects of perceptual quality and coherence.Furthermore, we include video-text consistency and video-image similarity from AIGCBench(Fan et al. (2023)), which offers a point of reference for comparing our benchmark with existing I2V evaluations.

## 5.3 QUANTITATIVE AND QUALITATIVE EVALUATION

Table 1 reports the quantitative comparison across four dimensions. (1) Closed-source models generally outperform open-source models.Wan2.1 leads among open-source models, CogVideoX slightly outperforms HunyuanVideo in reasoning, and Hailuo shows a clear advantage over SeedDance, es-

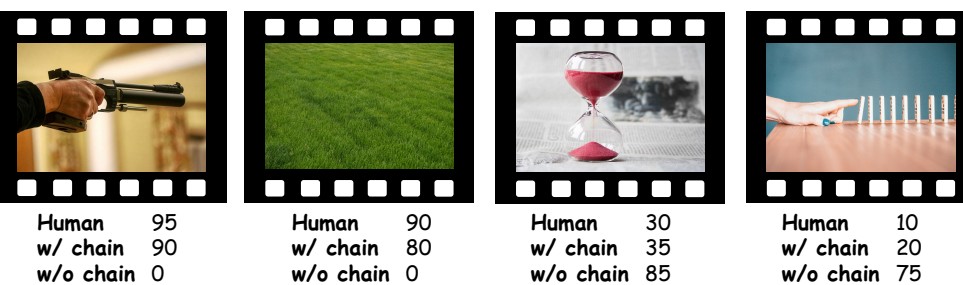

Figure 5: Evaluation on reasoning with Question Chain is aligned with human evaluation.

Table 3: Comparisons of human correlation among different dimensions.

| Metric | Spatial Understanding | | Attribute Binding | | Category Understanding | | Reasoning | | Avg. (ous) | |
|---|---|---|---|---|---|---|---|---|---|---|
| | $\tau$ | $\rho$ | $\tau$ | $\rho$ | $\tau$ | $\rho$ | $\tau$ | $\rho$ | $\tau$ | $\rho$ |
| Human Correlation | 0.6052 | 0.7345 | 0.2980 | 0.3566 | 0.5729 | 0.6683 | 0.6121 | 0.7329 | 0.5220 | 0.6231 |

pecially on the TC metric. (2) Their evaluation of generated video quality no longer meets the requirements with fundamental metrics only. For the semantic understanding dimension, it fails to accurately measure how well the generated output implements and responds to fine-grained information in the prompt. For the reasoning dimension, the quality of the generated result is no longer positively correlated with its alignment to the semantic information of the prompt. The qualitative evaluation shows similar observations.Hailuo exhibits strong reasoning capabilities. For example, given the prompt *"Leave the bananas for a week"*, the model can infer that the bananas will rot after some time,as shown in Appendix Figure 13.

## 5.4 HUMAN EVALUATION

We conduct a human evaluation of the 19 metrics across the four dimensions proposed in this paper. Specifically, we randomly select samples from our evaluation dataset, where each sample consists of a text prompt, an input image, and generated videos from different models. Evaluators are required to score each sample with simultaneous reference to the input image and text prompt, and each sample is evaluated by at least 8 evaluators. For each generated video, evaluators assign scores for the 19 metrics individually on a 1–5 scale. To reduce inter-evaluator bias, the scores from each user are further normalized before aggregation. Subsequently, we calculate the average score of all evaluators for each metric as the human subjective score.

We calculate Kendall's tau ($\tau$) and Spearman's rho ($\rho$) to reveal the similarity between our proposed metric and human evaluation, as shown in Table 3. Our proposed metrics exhibit high correlations with human judgments. In particular, the Spatial Understanding and Reasoning metrics achieve the strongest alignment with human evaluation, indicating their ability to capture semantic qualities and implicit reasoning beyond pixel-level measures. These results confirm that our benchmark provides reliable and human-consistent evaluation criteria.

## 6 CONCLUSIONS

We propose UI2V-Bench, a benchmark for evaluating I2V models in image understanding and prompt responsiveness. It covers semantic understanding (Spatial Understanding, Attribute Binding, and Category Understanding) and implicit reasoning, with automated evaluation pipelines based on MLLMs validated against human perception. We benchmark both open-source and commercial models through quantitative and qualitative analyses, revealing challenges in fine-grained subject–action alignment and in leveraging world knowledge for event prediction. We hope our work will inspire future improvements in the understanding ability of I2V models.

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

# A    QUESTION CHAIN DETAILS

This section presents the question chain score (Table 4) and specific question chain examples (Figure 6,7, 8). The score of the problem chain is basically consistent with the final score.

Table 4: Comparisons of Question Chain Score.

| Model | Reasoning | | | | |
|---|---|---|---|---|---|
| | Human Society | Natural Environment | Physical Interactions | Temporal Changes | Avg. |
| Wan2.1 | 57.08 | 52.01 | 45.11 | 27.38 | 46.37 |
| HunyuanVideo | 56.25 | 31.88 | 40.43 | 29.83 | 40.27 |
| CogvideoX | 52.15 | 43.94 | 38.64 | 32.65 | 42.35 |
| SeedDance | 56.62 | 62.45 | 73.03 | 39.00 | 58.02 |
| Hailuo | 79.64 | 68.11 | 65.17 | 73.62 | 69.89 |

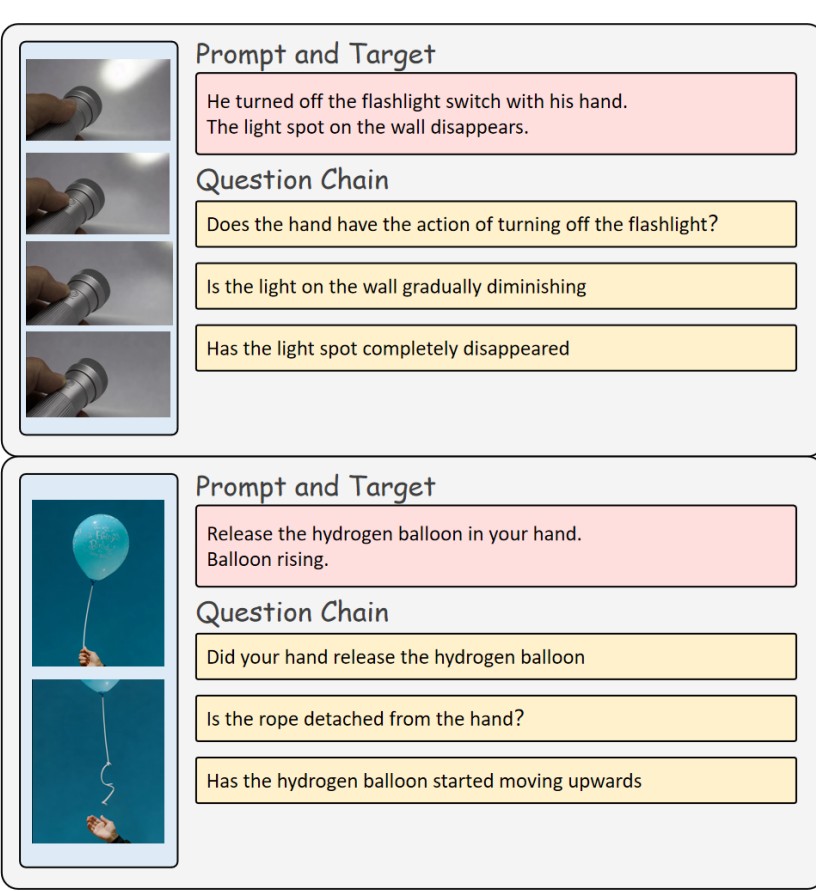

Figure 6: Question Chain cases

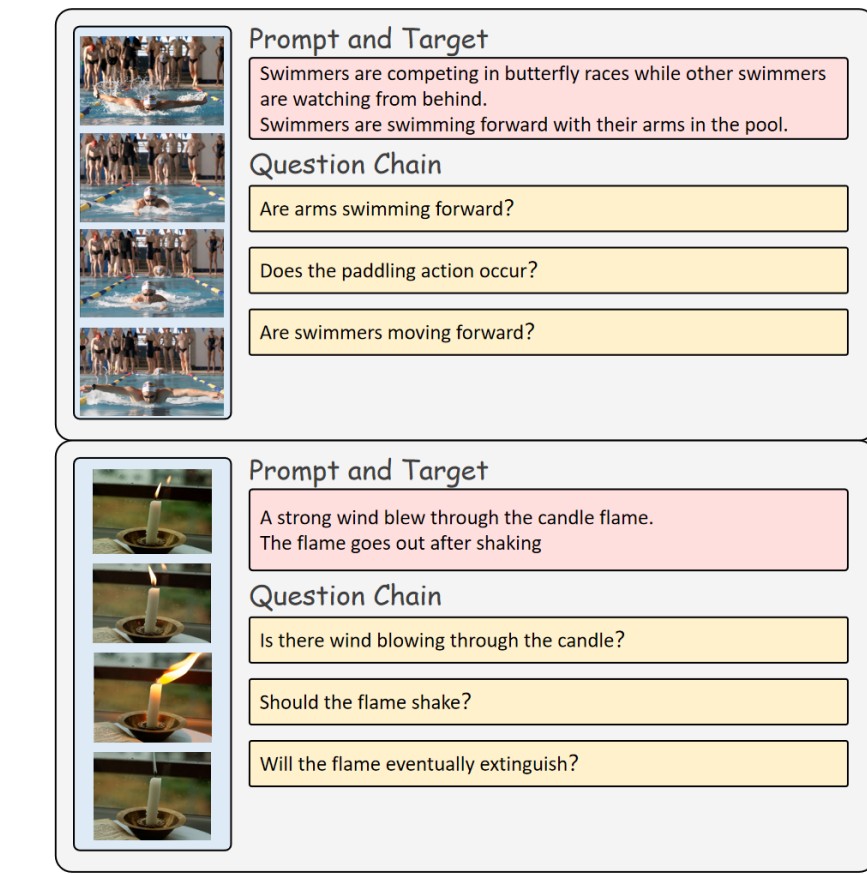

Figure 7: Question Chain cases

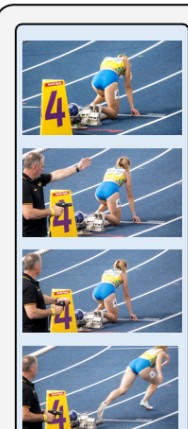

**Prompt and Target**

The referee fired the starting gun, signaling the start of the game. The athlete steps out of the starting block and completes the acceleration of the starting race.

**Question Chain**

Did the referee fire the starting gun?

Did the athlete start from the starting block?

Does the front leg stretch quickly, the back leg swing forward quickly?

Have you completed the starting acceleration?

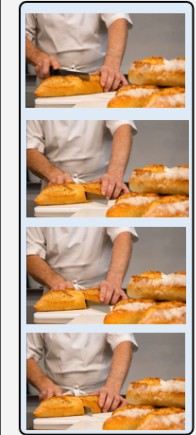

**Prompt and Target**

The chef placed a knife on the bread and wanted to cut it open. The chef holds a knife in his right hand and holds the bread with his left hand, cutting it open.

**Question Chain**

Is the chef holding a knife in his right hand?

Is the chef holding the bread with his left hand?

Is there any action of cutting bread?

Is the bread cut open?

Figure 8: Question Chain cases

## B  SYSTEM PROMPT

**Prompt Analyzer**

**Task Description**

You are a sentence parsing assistant. Please extract the subject and action from the input Chinese sentence and return it in JSON format.

**Prompt Analyze Rules**

**Subject:** Must include the noun and any preceding modifiers (if present).
  - Output the complete subject phrase, not just the core noun.
  - If there are any modifiers/qualifiers before the subject noun (e.g., adjectives, quantifiers, attributives, phrases, clauses, etc.), they must be retained.
  - Extract only the subject; do not include the predicate or object.

**Action:** Must include a complete description of the action.
  - If there are multiple subjects in the sentence, each subject should be paired with its corresponding action.
  - If the subject is not the agent of the action (e.g., "the largest fruit of that week"), it needs to be rewritten in passive voice.

The output format must strictly follow:
{"subject": ["..."], "action": ["..."]}
  - The content under subject and action should correspond sequentially by index.
  - Subject-action pairs with larger action amplitudes should be listed before those with smaller amplitudes.

Now process the following sentence: {Instruction}

**Score Generation**

**Task Description**

You are a helpful assistant and a brilliant action judger. You will receive a user-provided prompt [text1] and a subject description [text2].

**Output Judgment Result**

First, you need to analyze the subject description [text2] to determine whether the subject is being driven and rate the degree of the subject's movement (0-100, where 0 represents completely still and 100 represents intense movement) in the format: <target score> Output the target question completion score <animation score>.

Next, you need to judge the consistency of the action in the two given texts, focusing solely on the action. Similar semantics should be compromised. (1-100, where a higher score indicates greater consistency between the two given texts) in the format: <action consistency score> Output the target question completion score <action score>.

Figure 9: system prompt for semantic understanding evaluation

**Question Chain Generation**

**Task Description**

You are an expert in video generation evaluation. The text prompt for the video generation model is {prompt}, and the image is <image>. Please strictly follow the requirements below to generate a "step-by-step verification" question chain. Evaluate whether the video perfectly implements the following target text: {target}

**Question Generation Rules**

1. Each question must be based on clearly observable phenomena and allow for binary judgment (yes/no).
2. Questions should cover: trigger → feedback → result, and be output in sequence.
3. The language style should refer to the following example:
prompt = "A sniper pulls the trigger," target = "The bullet is fired."
Output
1. Does the hand show a motion of pulling the trigger?
2. Does the gun exhibit noticeable recoil?
3. Does the gun fire?
4. Is the bullet fired?

**Generate Output**

Now, for the given target text, generate a set of evaluation questions (3-5 questions).

**Score Generation**

**Task Description**

You are an expert in video event verification. You will receive a textual description of a video and a previously generated step-by-step verification question chain. Strictly follow the given question chain to verify whether the video description satisfies the final question (the target question).

**Verification Rules**

Verify each item sequentially according to the numbered order. Do not skip any steps.
Each judgment must be rigorously based on the content of the video description.Video Description{caption}
Question Chain{question}

**Output Judgment Result**

Determine the completion status of the target question and output a completion score for the target question (1–100, higher score indicates greater degree of completion) in the format: <target_score>Output the target question completion score<target_score>.
Determine the completion status of the question chain and output a completion score for the question chain (1–100, higher score indicates greater degree of completion) in the format: <chain_score>Output the question chain completion score<chain_score>.

Figure 10: system prompt for reasoning evaluation

# C    QUALITATIVE COMPARISON

**Prompt:** Man picking up unused paper

Figure 11: Qualitative Comparison on attribute.

**Prompt:** Take away the lettuce

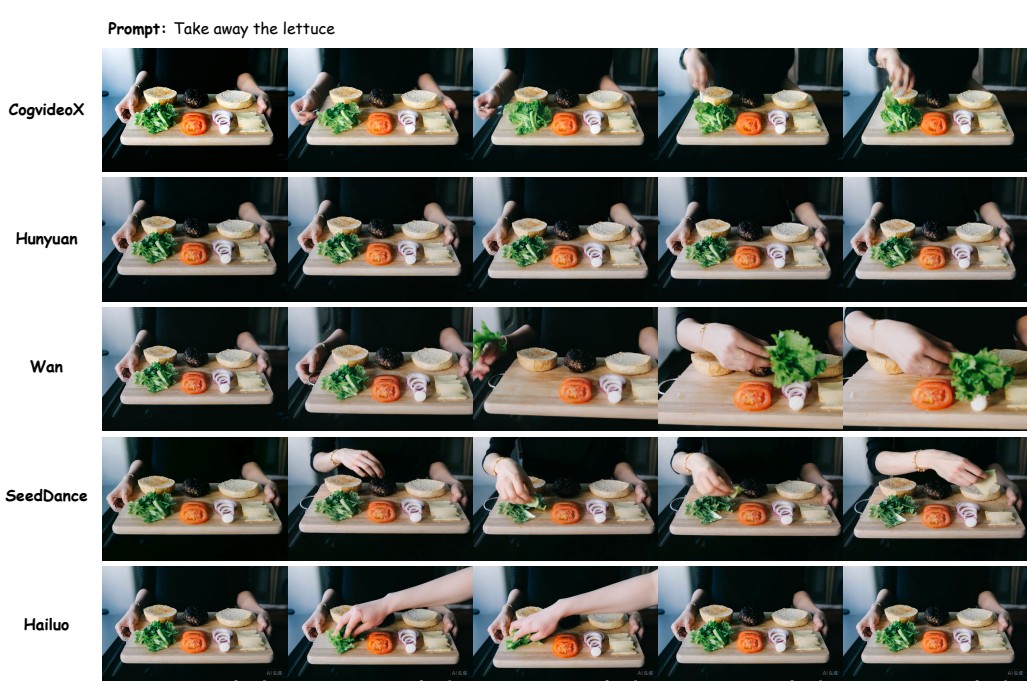

Figure 12: Qualitative Comparison on category.

**Prompt:** Leave the bananas for a week

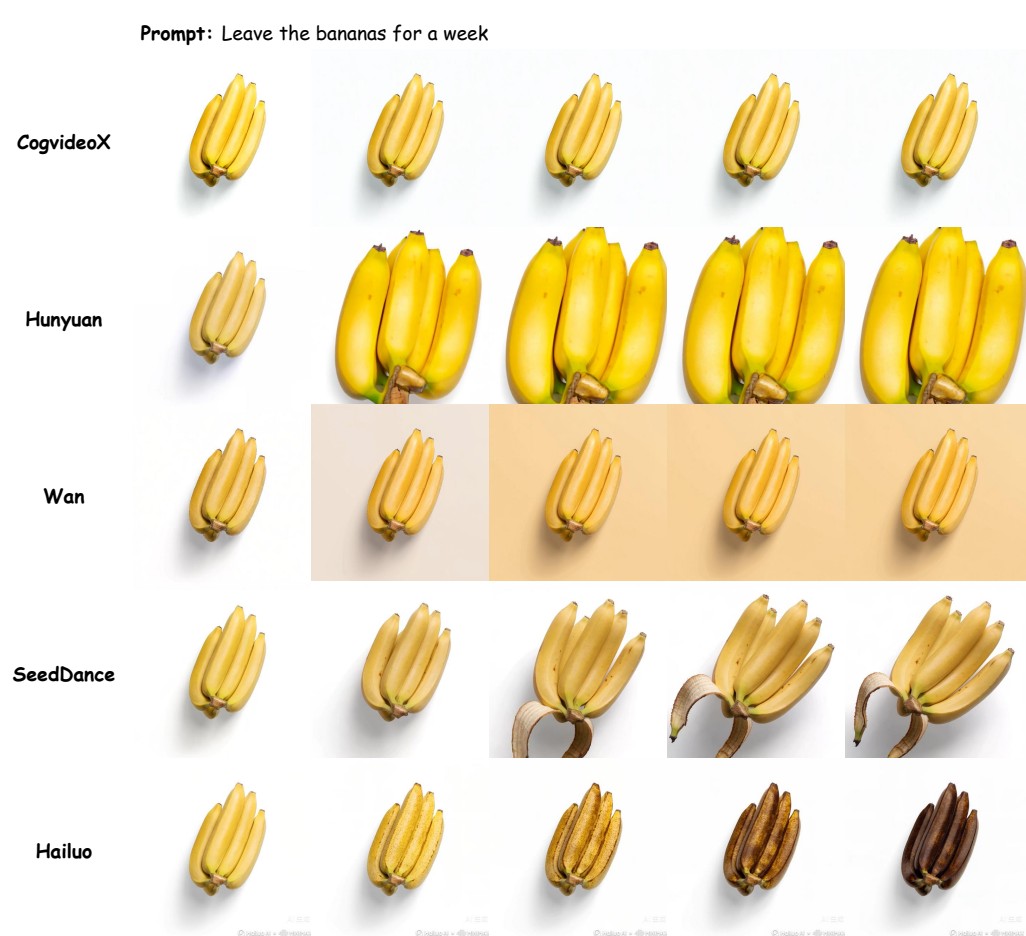

Figure 13: Qualitative Comparison on reasoning.

**Prompt:** The second person from left to right raises both hands

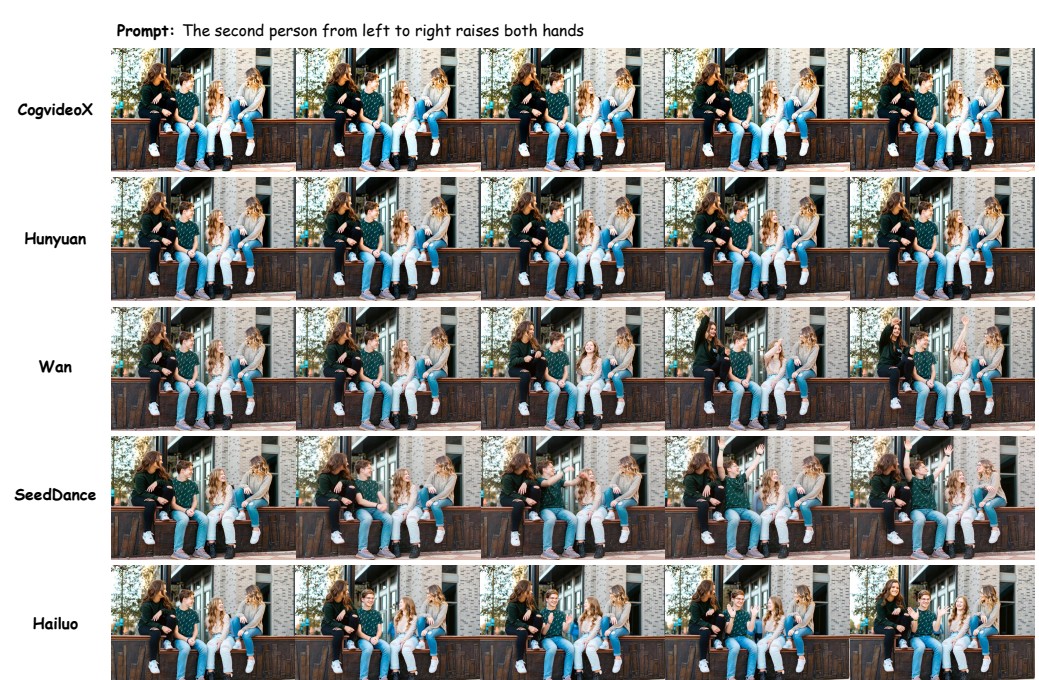

Figure 14: Qualitative Comparison on spatial.

