# OpenReview forum: "UI2V-Bench: An Understanding-based Image-to-video Generation Benchmark"
_ICLR.cc/2026/Conference — ICLR 2026 Conference Withdrawn Submission_

### Official Review · Reviewer_A2DE · 2025-10-31

**Soundness:** 3
**Presentation:** 3
**Contribution:** 2
**Rating:** 4
**Confidence:** 2

**Summary:**

This work introduces UI2V-Bench, a new benchmark designed to evaluate Image-to-Video generative diffusion models from the perspective of semantic understanding and reasoning, rather than only visual quality or temporal smoothness. The benchmark assesses four key capabilities and contain ~500 curated text–image pairs. Two evaluation pipelines based on Multimodal Large Language Models (MLLMs) are proposed: an instance-level pipeline for fine-grained semantic assessment and a feedback-based reasoning pipeline enabling step-by-step causal evaluation. Experiments on multiple open-source and commercial I2V models, along with human evaluation, confirm that the MLLM-based metrics correlate well with human judgment.

**Strengths:**

1. This work addresses an underexplored area—evaluating reasoning ability in image-to-video (I2V) generation. Existing benchmarks largely overlook semantic understanding and causal reasoning, making this contribution particularly valuable to the community.

2. Significant effort was invested in image collection and dataset construction, including sourcing high-quality images from the web to support a diverse set of evaluation scenarios.

**Weaknesses:**

1. Although the benchmark covers four dimensions (spatial understanding, attribute binding, category understanding, and reasoning) and 19 fine-grained sub-dimensions, the dataset size (~500 text–image pairs) may be insufficient considering the breadth of subcategories and reasoning types examined. Given the huge diversity and variability of real-world I2V scenarios, certain sub-dimensions (e.g., temporal changes, natural environment reasoning) are represented by only a small number of cases. This may constrain the benchmark’s ability to generalize and stress-test models under long-tail or compositionally complex situations.
2. While the benchmark introduces numerous sub-dimensions (e.g., emotion, object-holding, texture, etc.), the criteria distinguishing these subcategories are not always well formalized. Many examples remain simple or binary (e.g., “the blue ball rolls, the others stay still” for object attributes, or “birds with high positions fly away, birds with low positions stay still” for spatial understanding). Without clearer annotation guidelines or explicit difficulty levels, it is not obvious whether failure cases stem from misunderstanding of semantics or from prompt ambiguity.

**Questions:**

Refer to weakness

---

> ### Author Response · Authors · 2025-11-21
>
> Thank you for your valuable comments and acknowledgement of our novelty and effectiveness. We hope our response fully resolve your concerns.
>
> > Q1: the dataset size (~500 text–image pairs) may be insufficient considering the breadth of subcategories and reasoning types examined. Given the huge diversity and variability of real-world I2V scenarios, certain sub-dimensions (e.g., temporal changes, natural environment reasoning) are represented by only a small number of cases. This may constrain the benchmark’s ability to generalize and stress-test models under long-tail or compositionally complex situations.
>
> We are indeed continuously expanding the number of test cases, with the current plan to increase the total cases to over 1,000 and expand each sub-dimension to include more than 50 cases. The evaluation results in the paper will be updated accordingly. In future open-source releases, we will also continue to update and iterate the benchmark versions.
>
> > Q2: While the benchmark introduces numerous sub-dimensions (e.g., emotion, object-holding, texture, etc.), the criteria distinguishing these subcategories are not always well formalized.
>
> Actually, during the case construction process, we clearly defined the boundaries between different sub-categories and ensured mutual exclusivity of cases across sub-dimensions. Specifically, each case we constructed consists of an image and a prompt. While the same image can indeed be shared across multiple cases, the corresponding prompt definitively categorizes each case into a specific sub-dimension.
>
> For example, given an image containing two dogs with different color (e.g. a white dog and a black dog), this image can be assigned to different sub-categories through carefully designed prompts. Each prompt only describes the information necessary for the tested dimension to identify the specified subject.
>
> When the prompt states "the dog on the left runs away," it solely describes the spatial position ("left") of the designated dog, making this image-prompt pair belong to the left-right sub-dimension under the spatial dimension. Conversely, when the prompt states "the white dog runs away," it describes the color attribute ("white") of the specified dog, making this image-prompt pair fall under the color sub-dimension in attribute-binding.
>
> In summary, the construction and evaluation of each case ensure that the description of the specified subject only considers information from its respective dimension, thereby maintaining complete independence between different sub-dimensions.
>
> > Q3: Many examples remain simple or binary (e.g., “the blue ball rolls, the others stay still” for object attributes, or “birds with high positions fly away, birds with low positions stay still” for spatial understanding).
>
> To ensure that the evaluation focuses more on the core information of each sub-dimension itself, we strive to keep each prompt as concise and precise as possible. The general prompt tempate can be formed as: [Spatial/Attribute/Category description] + [Subject noun] + [An action the subject would naturally perform]. This design helps model concentrate on targeted dimension.
>
> In addition, in widely-used text-to-video benchmarks like VBench [1], the constructed prompts are also very succinct. In the color dimension, their cases are such as "a white bicycle" or "a red car". In comparison, our cases incorporate action information to evaluate whether the specified object in the image has been correctly driven.
>
> > Q4: Without clearer annotation guidelines or explicit difficulty levels, it is not obvious whether failure cases stem from misunderstanding of semantics or from prompt ambiguity.
>
> For both spatial and attribute dimensions, we ensured multiple subjects of the same category are present in each image, while guaranteeing that every image can be accurately interpreted solely through the information required by its sub-dimension. For example, in the color sub-dimension, the prompt "the white dog" ensures there is exactly one white dog in the image, thereby eliminating semantic ambiguity.
>
> The general structure of our prompts can be summarized as: [Spatial/Attribute/Category description] + [Subject noun] + [An action the subject would naturally perform]. All subjects used in the prompts are selected from commonly recognized categories.
>
> [1] VBench: Comprehensive Benchmark Suite for Video Generative Models, CVPR24.

---

### Official Review · Reviewer_hK7H · 2025-10-31

**Soundness:** 3
**Presentation:** 3
**Contribution:** 2
**Rating:** 4
**Confidence:** 3

**Summary:**

This paper introduces UI2V-Bench, a novel benchmark designed to evaluate Image-to-Video (I2V) generation models with a specific focus on semantic understanding and reasoning capabilities, aspects largely overlooked by existing benchmarks. The authors identify a critical gap where current I2V evaluations primarily focus on video quality and temporal consistency, failing to assess a model's ability to interpret image semantics, bind attributes, understand categories, or perform logical reasoning conforming to physical laws and human commonsense.

UI2V-Bench proposes four primary evaluation dimensions: spatial understanding, attribute binding, category understanding, and reasoning. These dimensions are further broken down into 9 aspects and 19 fine-grained sub-dimensions. To facilitate evaluation, the paper develops two Multimodal Large Language Model (MLLM)-based evaluation methods: an instance-level pipeline for fine-grained semantic understanding (spatial, attribute, category) and a feedback-based reasoning pipeline for step-by-step causal assessment. The benchmark includes approximately 500 carefully constructed text-image pairs and evaluates a range of both open-source and closed-source I2V models. Crucially, human evaluations are conducted, demonstrating strong alignment with the proposed MLLM-based metrics. The authors conclude that UI2V-Bench provides a robust framework and dataset to guide future research and development in I2V models by emphasizing comprehension and reasoning.

**Strengths:**

1. The paper successfully identifies and addresses a crucial limitation in current I2V evaluation, shifting the focus from purely perceptual quality to semantic understanding and reasoning. This is a significant step forward for the field. The introduction of four distinct dimensions (spatial, attribute, category, reasoning), further broken down into 19 sub-dimensions, offers a highly granular and comprehensive assessment of I2V model capabilities.
2. The two proposed MLLM-based evaluation pipelines (instance-level and feedback-based) are creative and leverage the strengths of modern MLLMs for automated, fine-grained assessment, which is essential for scalability. The feedback-based reasoning pipeline is particularly novel in guiding MLLMs through causal inference.
3. The validation of MLLM-based metrics against human judgment, showing high correlations (Kendall's tau and Spearman's rho), significantly bolsters the credibility and reliability of UI2V-Bench. The benchmark evaluates both open-source and commercial I2V models, providing insights into the current state of the art and highlighting areas for improvement. The dataset of ~500 text-image pairs is tailored to specific challenges. The paper clearly describes the construction of the benchmark, the evaluation pipelines, and the experimental setup, making it easy for other researchers to understand and potentially build upon.

**Weaknesses:**

1. While "carefully constructed," a dataset of approximately 500 text-image pairs, when distributed across 4 main dimensions and 19 sub-dimensions, might be considered relatively small. This could limit the diversity and statistical robustness for some niche scenarios within each sub-dimension, potentially leading to an incomplete picture of a model's true capabilities across all possible variations.
2. The heavy reliance on MLLMs for evaluation, while validated, introduces a dependency on the capabilities and potential biases of these underlying models. The paper mentions "textual biases" (line 349) but could further elaborate on how the feedback-based pipeline fully mitigates this, or if there are inherent limitations that still exist due to MLLM behavior. The performance of the benchmark itself is tied to the evolving capabilities of MLLMs.
3. While system prompts are provided, more detailed examples or a rubric for how the "Result Judge" (for instance-level) and the final scoring (for reasoning) precisely convert MLLM outputs/responses into a quantitative score would enhance transparency and reproducibility. For instance, what constitutes a "correct" comparison or a "successful" chain of questions leading to a score?
4. Running evaluations using multiple advanced MLLMs and specialized video-LLMs (VideoRefer, Tarsier) could be computationally intensive and incur significant API costs for closed-source MLLMs. This might pose a barrier for researchers with limited resources who wish to use the benchmark extensively.

**Questions:**

1. Could the authors elaborate on the process of constructing the ~500 text-image pairs? What specific strategies were employed to ensure diversity and representativeness across all 19 sub-dimensions, especially given the relatively small total number of samples? Were there any specific challenges in creating complex reasoning cases?
2. The paper mentions "textual biases" for MLLMs. Could the authors provide a more in-depth discussion on these biases and how the feedback-based reasoning pipeline specifically addresses or mitigates them beyond just generating a chain of questions? Are there known failure modes or types of reasoning where current MLLMs still struggle, even with this feedback mechanism?
3. What is the estimated computational cost (e.g., GPU hours, API calls for commercial MLLMs) for running a full evaluation of a single I2V model across the entire UI2V-Bench? This information would be very valuable for researchers planning to utilize the benchmark.
4. For the "Result Judge" component in the instance-level evaluation and the final score aggregation in the feedback-based reasoning, could the authors provide more concrete examples or a detailed rubric of how MLLM outputs are translated into quantitative scores? This would improve the transparency and reproducibility of the evaluation process.
5. Are there plans to expand the dataset size in future iterations of UI2V-Bench, particularly for sub-dimensions that might have fewer examples, to further enhance its statistical robustness and coverage?

---

> ### Author Response · Authors · 2025-11-21
>
> > Q1: Could the authors elaborate on the process of constructing the ~500 text-image pairs?
>
> Regarding the input images selection:
> For the semantic dimension, we filtered images from Unsplash and Pexels.
>
> - For the spatial dimension, an image should contain 2–4 subjects of the same category with strictly defined spatial relationships.
>
> - For the attribute dimension, we first ensure that all the subjects belonging to the same category. Then we further guarantee that at least one subject in the image is unique under the specific attribute. For example, in the color sub-dimension, it is required that at least one subject in the image has a yellow color, thereby allowing the target subject to be uniquely identified through the description of its color.
>
> - For the category dimension, the images contain objects of multiple categories.
>
> For the reasoning dimension, we first constructed the prompts and then selected matching images.
>
> As for the prompt construction:
>
> - For sematic dimension, prompt are carefully designed according to images. They can generally be summarized as: [Spatial/Attribute/Category description] + [Subject noun] + [An action the subject would naturally perform]. We ensure that the prompt only drives one object or person.
>
> - For reasoning dimension, the prompts no longer describe a specific subject to drive its motion. Instead, they provide a precondition for the input image as the 'cause ' guiding the I2V model to generate the corresponding 'effect'.
>
> > Q2: What specific strategies were employed to ensure diversity and representativeness across all 19 sub-dimensions? Were there any specific challenges in creating complex reasoning cases?
>
> To ensure diversity and representativeness across all 19 sub-dimensions, we are currently collecting source images that contain most categories mentioned in the MSCOCO dataset, and we will also expand to include more common categories in the final version of the benchmark.
>
> The main challenge we encountered is that for some dimensions, it is difficult to find suitable images in natural scenes, as real-life scenarios rarely contain multiple subjects with distinct attributes simultaneously. To address this, we have also utilized GPT-assisted generation to create image-text pairs that meet the requirements and supplement the case collection.
>
> > Q3: The paper mentions "textual biases" for MLLMs. Could the authors provide a more in-depth discussion on these biases and how the feedback-based reasoning pipeline specifically addresses or mitigates them beyond just generating a chain of questions?
>
> We first apologize for the misunderstanding. The feedback-based reasoning pipeline cannot address "textual biases" within MLLMs. To overcome this, we avoid direct cross-modal comparison by converting all relevant video information into textual representations.
>
> For example, when an input image shows a hand holding a balloon with the prompt "release the balloon". If directly querying the video understanding model with "Was the balloon released?" it may yield incorrect judgments. However, by first converting to textual descriptions for analysis, accurate assessment can be achieved. This evaluation methodology avoiding MLLM cross-modal hallucination has also proven effective in Video-Bench [1].
>
> > Q4: Computational cost.
>
> - For the semantic understanding dimensions, we employ an instance-level evaluation framework. On average, evaluating a single case takes 3.66 s, which is broken down as follows: Prompt Analyzer requires 0.45 s, Subject Segmentor 0.43s, Action Descriptor 2.63 s, and Prompt Judger 0.15 s.
>
> - For the reasoning dimension, the feedback-based pipeline takes an average of 4.14 s per case. This process consists of two main stages: video description extraction (1.49 s) and assessment based on a Question Chain (2.65 s).
>
> All the evaluation experiments run on H100. MLLM uses the locally deployed Qwen2.5-VL-72B-Instruct. The inference speed is acceptable.
>
> > Q5: Could the authors provide more concrete examples or a detailed rubric of how MLLM outputs are translated into quantitative scores?
>
> Actually, we show the details in Appendix. The system prompt presented in Section B elaborates on the specific scoring details for the MLLM. Besides,  Section A introduces question chain detail for  feedback-based reasoning pipeline.
>
> > Q6: Are there plans to expand the dataset size in future iterations of UI2V-Bench, particularly for sub-dimensions that might have fewer examples, to further enhance its statistical robustness and coverage?
>
> We are indeed continuously expanding the number of test cases, with the current plan to increase the total cases to over 1,000 and expand each sub-dimension to include more than 50 cases. The evaluation results in the paper will be updated accordingly in a few days. In future open-source releases, we will also continue to update and iterate the benchmark versions.
>
> [1] Video-Bench: Human-Aligned Video Generation Benchmark, CVPR25.

---

### Official Review · Reviewer_9dyM · 2025-11-01

**Soundness:** 2
**Presentation:** 2
**Contribution:** 3
**Rating:** 4
**Confidence:** 4

**Summary:**

The paper proposes a new benchmark for image-to-video generation that tests whether a model can reason about plausible events. The evaluation metrics span four perspectives: spatial understanding, attribute binding, category understanding, and reasoning. The judge is an MLLM-driven agent system.

**Strengths:**

1. The paper is well-written with a clear storyline. The structure of the paper is well designed.
2. The research topic of I2V evaluation is timely, important, and motivated.
3. The evaluation metrics cover sufficiently many perspectives.

**Weaknesses:**

1. The heavy dependency on using MLLMs as judges can be inaccurate. I understand that the paper already has a few user studies (Section 5.4 human evaluation), but that is relatively vaguely described. What are the qualifications of the selected users? How does the author team guarantee that the users are sufficiently independent to the authors?
2. More principled video evaluation metrics that utilize symbolic structures will be more convincing (e.g., the consistency and dynamics metrics presented in WorldScore (ICCV 2025)).

**Questions:**

The weakness is listed above, and my initial rating for the paper is borderline rejection (4). My final rating will be conditioned on the soundness of authors' rebuttal.

---

> ### Author Response · Authors · 2025-11-21
>
> We are grateful for your positive review and valuable comments, and we hope our response fully address your concerns.
>
> > Q1: What are the qualifications of the selected users? How does the author team guarantee that the users are sufficiently independent to the authors?
>
> - Ensuring Independence and Objectivity: To guarantee sufficient independence, none of the participants are included in the author team. Participants remain unaware of the source of the generated videos for each case. Additionally, videos from different models are randomly arranged on the scoring interface. This setup ensures absolute objectivity by avoiding potential bias toward any specific model.
>
> - Ensuring Accuracy and Expertise: All selected participants possess expertise in the field of video understanding and generation (e.g., photography enthusiasts, AI/data engineers), guaranteeing the professionalism of the scoring.

---

### Official Review · Reviewer_fmss · 2025-11-03

**Soundness:** 2
**Presentation:** 3
**Contribution:** 2
**Rating:** 4
**Confidence:** 3

**Summary:**

This paper introduces U12V-Bench, a new benchmark for evaluating Image-to-Video (I2V) models. It focuses on a model's ability to understand the input image and perform reasoning, rather than just assessing low-level video quality. The benchmark tests four key dimensions: Spatial Understanding, Attribute Binding, Category Understanding, and Reasoning. It uses innovative MLLM-based methods for automated evaluation, and the results show a strong correlation with human judgment.

**Strengths:**

The main strength is its novel and important focus on semantic understanding and reasoning, which addresses a clear gap in existing I2V evaluation. The four evaluation dimensions are well-chosen and comprehensive. The two proposed evaluation pipelines are clever; the instance-level pipeline allows for fine-grained checks, and the feedback-based "question chain" method for reasoning is particularly effective. The high correlation with human evaluation strongly validates their automated metrics. The planned release of the dataset is a valuable contribution to the community.

**Weaknesses:**

The current dataset size of ~500 samples is a good start but could be expanded in the future for even better coverage and robustness. The evaluation pipelines, especially the one involving segmentation and video LLMs, seem computationally heavy. It would be helpful to briefly discuss the computational cost or efficiency. There is a minor terminology inconsistency in the "Attribute Binding" dimension between "Textures" in the table and "material" in the text.

**Questions:**

1. Do terms "Textures" (in Table 1) and "material" (in Section 3.1.2) refer to the same concept?

2. Do authors have a specific timeline for the public release of the benchmark dataset and evaluation code?

3. Could authors provide a rough estimate of the computational cost required to run the full evaluation suite on a single model?

---

> ### Author Response · Authors · 2025-11-21
>
> Thank you for your valuable comments and acknowledgement of our effectiveness. We hope our response fully resolve your concerns.
>
> > Q1: Do terms "Textures" (in Table 1) and "material" (in Section 3.1.2) refer to the same concept?
>
> Thank you for your reminder. We apologize for the carelessness in the terminology inconsistency. The terms "Textures" (in Table 1) and "material" (in Section 3.1.2) both refer to the same concept. We will update the manuscript and standardize all relevant instances to use "material" throughout the manuscript.
>
> > Q2: Do authors have a specific timeline for the public release of the benchmark dataset and evaluation code?
>
> The open-sourcing of our evaluation dataset and assessment code has been steadily progressing. Currently, the related code organization, documentation, and data cleaning efforts are nearing completion.
>
> We will release the evaluation code and the first 500 cases of UI2V-Bench in December this year, and the expanded cases will be released one month later.
>
> We appreciate your reminder, which helps ensure our work can make a tangible contribution to the community.
>
> > Q3: Could authors provide a rough estimate of the computational cost required to run the full evaluation suite on a single model?
>
> Our evaluation pipeline includes two different frameworks, which are carefully designed for different dimensions . For the semantic understanding dimensions, we employ an instance-level evaluation framework. On average, evaluating a single case takes 3.66 seconds, which is broken down as follows: the prompt analyzer requires 0.45 s, the subject segmentor 0.43s, the action descriptor 2.63 s, and the prompt judger 0.15 s.
>
> For the reasoning dimension, we utilize a feedback-based pipeline. Evaluating a single case takes an average of 4.14 s. This process consists of two main stages: video description extraction (1.49 s) and assessment based on a Question Chain (2.65 s).
>
> All the evaluation experiments run on H100. MLLM uses the locally deployed Qwen2.5-VL-72B-Instruct. The inference speed is acceptable.
>
> > Q4: The current dataset size of ~500 samples is a good start but could be expanded in the future for even better coverage and robustness.
>
> Thank you for your suggestion. We are indeed continuously expanding the number of test cases, with the current plan to increase the total cases to over 1,000 and expand each sub-dimension to include more than 50 cases. The evaluation results in the paper will be updated accordingly. In future open-source releases, we will also continue to update and iterate the benchmark versions.

---

### Note · Authors · 2026-01-28

I have read and agree with the venue's withdrawal policy on behalf of myself and my co-authors.

---

### Meta-Review · Area_Chair_Z8mg · 2026-01-05

**Summary:**

This paper introduces UI2V-Bench, a new benchmark for evaluating Image-to-Video (I2V) models with a focus on semantic understanding and reasoning capabilities.

All reviewers agree that the motivation of this work is strong. However, two main concerns have been raised by the reviewers. 1) Reviewers fmss, hK7H, and A2DE all raised concerns that the dataset size is too small, particularly when divided across 4 dimensions and 19 sub-dimensions. 2) Reviewers 9dyM and hK7H raised concerns regarding the heavy reliance on MLLMs as judges.

After considering the paper, the reviews, and the rebuttal, the AC agrees with all the reviewers that the dataset scale is limited and the evaluation is heavily dependent on MLLM. The current submission does not yet provide sufficient evidence that the benchmark is comprehensive and reliable. Therefore, the AC recommends Reject.

**Reviewer Concerns:**

The rebuttal does address several questions and supplement implementation details, but the two main concerns shared across reviewers remain largely outstanding, i.e., 1) dataset size and coverage, 2) heavy reliance on MLLMs as judges.

**Reviewer Scores:**

Since the rebuttal mainly focuses on promises of future work rather than addressing the limitations of 1) dataset size and coverage, 2) heavy reliance on MLLMs as judges, the reviewers would likely keep their initial rating.

---

### Decision · Program_Chairs · 2026-01-26

Reject